# Synchronization of Coupled Delayed Discontinuous Systems via Event-Trigged Intermittent Control

1st Rongqiang Tang
*College of Electronics and Information Engineering*
*Sichuan University*
Chengdu, Sichuan
tangrongqiang@stu.scu.edu.cn

2nd Xinsong Yang*
*College of Electronics and Information Engineering*
*Sichuan University*
Chengdu, Sichuan
xinsongyang@scu.edu.cn

*Abstract*—This talk focuses on the complete synchronization of coupled delayed discontinuous systems (DDSs). Without constraints on the derivatives of time delays, several new conditions are exploited to guarantee the global existence of Filippov solutions for DDSs. A nonsmooth intermittent control combined with an event-triggering strategy is then designed. The conspicuous feature of this control scheme is that the measurement error in the event-triggering mechanism is formulated as a linear form, which can reduce computation burden compared to classical approaches. To address the challenges posed by Filippov solutions and intermittent control, novel analytical techniques, including an original lemma and a weighted-norm-based Lyapunov function, are developed so that sufficient synchronization conditions for DDSs are obtained. Finally, the effectiveness of the theoretical findings is confirmed by Hopfield neural networks.

*Index Terms*—Discontinuous systems, event-triggered intermittent control, Filippov solution, synchronization, time delays.

## I. INTRODUCTION

Coupled discontinuous systems (DSs), modeled by some interconnected differential equations with discontinuous right-hand sides, are a special type of complex network. Their applications span various areas of applied science and engineering, such as variable structure systems, neural networks [1], control synthesis [2], etc. Recently, there has been substantial attention on the dynamic behaviors of DSs with or without time delays, covering stability, stabilization, and synchronization [3]–[5].

Considering the discontinuities of the states on the right-hand side of DSs, especially delayed DSs (DDSs), it is paramount to discuss the existence of Filippov solutions. Some limitations on time delays are necessary to ensure the existence of Filippov solutions for DDSs. For example, literature [1] considered DDSs with constant delays. Liu et al. [6] demanded that the state variables with time delays satisfy $\|z(t - \sigma(t))\| \leq \|z(t)\| + \max_{1 \leq i \leq n} \max_{-\sigma \leq s \leq 0} \{z_i(s)\}$, where $z(t) \in \mathbb{R}^n$ is the state variable and $\sigma(t) \in [0, \sigma]$ is the time delay. Yang et al. [7], [8] provided sufficient criteria for

This work was supported in part by the National Natural Science Foundation of China (NSFC) under Grant Nos. 62373262 and 62303336, and in part by the Central guiding local science and technology development special project of Sichuan, and in part by the Fundamental Research Funds for Central Universities under Grant No. 2022SCU12009, and in part by the Sichuan Province Natural Science Foundation of China (NSFSC) under Grant Nos. 2022NSFSC0541, 2022NSFSC0875, 2023NSFSC1433.(*Corresponding Author: Xinsong Yang*)

the existence of global Filippov solutions for DDSs, based on the condition that the derivatives of time delays are less than 1. However, in reality, the derivatives of some time delays can exceed or equal 1, and even be non-differentiable in some cases. A fundamental question arises: What conditions guarantee the existence of Filippov solutions for DDSs when these constraints are removed?

To study the synchronization of coupled DDSs (CDDSs), the basic idea is to transform CDDSs into uncertain systems using Filippov regularization and the measurable selection theorem, and then to address the corresponding issues for the uncertain systems [8]. Quasi-synchronization criteria for CDDSs have been obtained via smooth state feedback control [6], [9]. A nonsmooth control incorporating sign functions was proposed to achieve complete synchronization of CDDSs [7], where the sign function is use to mitigate the effects of uncertainties caused by Filippov solutions. Subsequent results on exponential, finite-time, and fixed-time synchronization of CDDSs have been published in [10]–[13]. However, little work has been done to achieve the complete synchronization of CDDSs via intermittent control. Actually, intermittent control offers better robustness and lower control cost than continuous control, as control signals can be artificially interrupted without affecting the final control purposes [14]–[18]. If the intermittent control is adopted for complete synchronization of CDDSs, the main obstacle lies in that the uncertainties posed by Filippov solution are difficult to cancel out during the interrupted intervals of control signals. So, how to develop new analytical methods to study the complete synchronization of CDDSs with intermittent control is another motivation.

Event-triggered control has recently sparked increasing interest due to its ability to reduce computational overhead by updating the sampled signal based on a preset supervision mechanism [19]–[21]. To fully leverage the merits of event-triggered strategy and intermittent control, this paper considers the complete synchronization of general CDDSs via a novel event-trigged intermittent control. The primary contributions of this work are:

1) The existence of Filippov solutions of DDSs is discussed. Different from existing papers [1], [6]–[8], several harsh constrictions on delays are removed.

2) A novel lemma is developed to address the difficulties

induced by intermittent control. Then, complete synchronization criteria for CDDSs with intermittent control are obtained for the first time.

3) A simple robust intermittent control scheme is designed by combining an event-triggered strategy with nonsmooth control. Unlike many event-triggered nonsmooth controls [12], [17], the measurement error (ME) in a linear form for the event-triggering mechanism (ETM) is considered, which facilitates easy computation (see Table I).

**Notation:** Let $\mathcal{D}^+[\cdot]$ be the upper right Dini derivative operator. $\mathbb{N}_k^j \triangleq \{k, k+1, \ldots, j\}$ with $k < j \in \mathbb{N}$, $\mathsf{dg}(\cdot)$ is the block-diagonal matrix. For $a \in \mathbb{R}^n$, let $\mathrm{cl}(a_i)_n = (a_1, a_2, \ldots, a_n)^\mathsf{T}$, and $\mathrm{dg}(a_i)_n = \mathrm{diag}(a_1, a_2, \ldots, a_n)$, $\mathsf{sg}(a) = \frac{a}{\|a\|}, \|a\| \neq 0$, otherwise $\mathsf{sg}(a) = 0$. The other notations used in this paper are same as those in [16].

## II. PRELIMINARIES

In this paper, the problem of synchronization and control in an array of coupled DDSs is considered. Before starting the research works, several necessary preparations on the solution of DDSs and stability theorem are provided.

### A. Filippov solution of DDSs

Consider a DDS as follows:

$$\dot{z}(t) = F(z, z_\sigma), \ z(o) = \tau(o) \in \mathcal{C}([-\sigma, 0], \mathbb{R}^n). \quad (1)$$

Here $F(z, z_\sigma) \triangleq Cz(t) + Ah(z(t)) + Bg(z(t - \sigma(t)))$, $z(t) \in \mathbb{R}^n$ denotes the state vector, $\sigma(t) \in [0, \sigma]$ is the bounded delay, $C, A = (a_{ij})_{n \times n}$, and $B = (b_{ij})_{n \times n} \in \mathbb{R}^{n \times n}$ are known constant matrices, nonlinear functions $h(\cdot), g(\cdot) : \mathbb{R}^n \to \mathbb{R}^n$ are continuous except on a series of smooth hypersurfaces domains [7]. Chosen an initial value $z(o)$ for system (1), its trajectory can establish the desired state, such as equilibrium point, chaotic orbit, or nontrivial periodic orbit.

Due to the discontinuity of $\mathrm{a}(\cdot)$ with $\mathrm{a} = \{h, g\}$, classical solutions of DDS (1) do not exist. To further study the dynamical behaviors of DDS (1), this paper utilizes the framework of the Filippov solution, in which the definition of Filippov solution can be founded in [6]–[8]. It is concluded that, for DDS (1), there exists a continuous function $z(t)$ on $[-\sigma, \mathfrak{t}]$ to be absolutely continuous on $[0, \mathfrak{t}]$ such that

$$\dot{z}(t) = \mathbb{F}(z, \gamma, \zeta_\sigma), \ a.a. \ t \in [0, \mathfrak{t}], \quad (2)$$

where $\mathbb{F}(z, \gamma, \zeta_\sigma) = Cz(t) + A\gamma(t) + B\zeta(t - \sigma(t))$, $\gamma(t) \in \mathsf{F}\{h(z(t))\}$ and $\zeta(t - \sigma(t)) \in \mathsf{F}\{g(z(t - \sigma(t)))\}$ are measurable functions, and $\mathsf{F}\{\cdot\}$ is the Filippov set-valued map [22].

For the Cauchy problem of DDS (1) in the sense of Filippov, it implies that there is a triple of function $(z(t), \gamma(t), \zeta(t))$ : $[-\sigma, \mathfrak{t}] \to \mathbb{R}^n \times \mathbb{R}^n \times \mathbb{R}^n$ such that $z(t)$ is a Filippov solution on $[-\sigma, \mathfrak{t}]$ with $\mathfrak{t} > 0$ and

$$\begin{cases} \dot{z}(t) = \mathbb{F}(z, \gamma, \zeta_\sigma), \ a.a. \ t \in [0, \mathfrak{t}], \\ \gamma(s) = \zeta(s) = \mathsf{F}\{\phi(s)\}, \ a.a. \ s \in [-\sigma, 0], \\ z(s) = \varphi(s), \ \forall s \in [-\sigma, 0], \end{cases} \quad (3)$$

where $\varphi(t)$ is a continuous function on $[-\sigma, 0]$ and $\phi(t)$ is a measurable selection function.

The following lemma provides some mild conditions to ensure the existence of Filippov solutions for DDS (1).

*Lemma 1:* Suppose that $\mathrm{a}(0) = 0, \mathrm{a} = \{h, g\}$ and there exist constants $d_{rj}^{\mathrm{a}} \geq 0$ and $\widehat{d_r^{\mathrm{a}}} \geq 0$ such that, for $\forall \mathsf{x} = \mathrm{cl}(x_i)_n, \mathsf{y} = \mathrm{cl}(y_i)_n \in \mathbb{R}^n$,
$(\mathbf{A}_1) : |\mathrm{a}_r(\mathsf{x}) - \mathrm{a}_r(\mathsf{y})| \leq \sum_{j=1}^n d_{rj}^{\mathrm{a}} |x_j - y_j| + \widehat{d_r^{\mathrm{a}}}, \ r \in \mathbb{N}_1^n$.
Then, there is at least one Filippov solution $z(t)$ to DDS (1) on $[0, +\infty)$.

*Proof:* The prove process is similar to those in [7], [8] with slightly changes, that is, the Cauchy problem in (3) is transformed into a fixed point problem.

Denote a map $\mathbb{G}(z) : \mathcal{C}([-\sigma, \mathfrak{t}], \mathbb{R}^n) \to \mathcal{C}([-\sigma, \mathfrak{t}], \mathbb{R}^n)^1$ as:

$$\mathbb{G}(z) = \begin{cases} e^{Ct} z(0) + \int_0^t e^{C(t-s)} \big[ B\mathsf{F}\{g(z(t - \sigma(t)))\} \\ \quad + A\mathsf{F}\{h(z(t))\} \big] \mathrm{d}s, \ t \in [0, \mathfrak{t}], \mathfrak{t} > 0, \\ \varphi(s), \forall s \leq 0. \end{cases} \quad (4)$$

It has that $\mathbb{G}(z)$ is completely continuous and upper semi-continuous with convex closed values. Further, one knows that the solutions of the Cauchy problem of DDS (3) are the fixed points of $\mathbb{G}(z)$.

By $(\mathbf{A}_1)$, the set $\Omega = \{z \in \mathcal{C}([-\sigma, \mathfrak{t}], \mathbb{R}^n) : \lambda z \in \mathbb{G}(z), \lambda > 1\}$ is non-empty. Next, let us prove that the set $\Omega$ is bounded.

For $z \in \Omega$, it holds that $\lambda z \in \mathbb{G}(z)$ for $\lambda > 1$. So, there are $\gamma(t) \in \mathsf{F}\{h(z(t))\}$ and $\zeta(t - \sigma(t)) \in \mathsf{F}\{g(z(t - \sigma(t)))\}$ such that

$$z(t) = \frac{1}{\lambda} \Big[ z(0) e^{Ct} + \int_0^t e^{C(t-s)} \mathbb{c}(s) \mathrm{d}s \Big], \ a.a. \ t \in [0, \mathfrak{t}], \quad (5)$$

where $\mathbb{c}(t) = A\gamma(s) + B\zeta(s - \tau(s))$.

In view of $(\mathbf{A}_1)$, there are constants $D_{\mathrm{a}}$ and $d_{\mathrm{a}}$ such that

$$\|\mathbb{c}(t)\| \leq D_h \|A\| \|z(t)\| + D_g \|B\| \|z(t - \sigma(t))\| + \mathbb{d}, \quad (6)$$

where $\mathbb{d} = (d_h \|A\| + d_g \|B\|)$ and $\mathrm{a} = \{h, g\}$. Considering inequalities (5) and (6), it follows that

$$\|z(t)\| \leq e^{\|C\| t} \Big[ \mathbb{y}(t) + D_g \|B\| \int_0^t e^{-\|C\| s} \|z(s - \tau(s))\| \mathrm{d}s$$
$$+ D_h \|A\| \int_0^t e^{-\|C\| s} \|z(s)\| \mathrm{d}s \Big], \ a.a. \ t \in [0, \mathfrak{t}],$$

which implies that

$$\mathbb{z}(t) \leq \mathbb{y}(t) + \mathcal{D} \int_0^t \mathbb{z}(s) \mathrm{d}s, \ a.a. \ t \in [0, \mathfrak{t}], \quad (7)$$

where $\mathbb{z}(t) = e^{-\|C\| t} \sup_{\theta \in [-\sigma, t]} \|z(\theta)\|$, $\mathcal{D} = (D_h \|A\| + D_g \|B\|)$, and $\mathbb{y}(t) = \|z(0)\| + \frac{\mathbb{d}}{\|C\|} (1 - e^{-\|C\| t})$.

Note that, it is easy to obtain $y_{\max} = \|z(0)\| + \frac{\mathbb{d}}{\|C\|}$ is a upper bound of $\mathbb{y}(t)$ on $[0, +\infty)$. Then, from inequality (7) and the Gronwall's lemma, it has

$$e^{-\|C\| t} \|z(t)\| \leq \mathbb{z}(t) \leq y_{\max} e^{\mathcal{D} t}, a.a. \ t \in [0, \mathfrak{t}], \quad (8)$$

---

[1]$\mathcal{C}([-\sigma, \mathfrak{t}], \mathbb{R}^n)$ is the Banach space of the $n$-dimensional vector-valued continuous functions defined on $[-\sigma, \mathfrak{t}]$ with norm defined by $\|x\|_\infty = \sup\{\|x(t)\|, t \in [-\sigma, \mathfrak{t}]\}$.

which further means that $\Omega$ is bounded, a.a. $t \in [-\sigma, \mathfrak{t}]$.

From the discussions in [7], it is deduced that $\mathbb{G}(z)$ has a fixed point for $\forall t > 0$, which infers that a Filippov solution to DDS (1) can be defined on $[0, +\infty)$. ∎

*Remark 1:* Delay $\sigma(t)$ in DDS (1) is merely bounded, which is a milder condition than those in [1], [7], [8]. For instance, the existence of Filippov solutions for DDSs has been discussed in [1], [7], [8] under the condition that the derivatives of delays are differentiable and their values do not exceed 1. Moreover, the proof in Lemma 1 differs from that in [6]. The technique in [6] for handling time delay involves the inequality $\|z(t - \sigma(t))\| \leq \max\limits_{1 \leq i \leq n} \max\limits_{-\sigma \leq s \leq 0} \{z_i(s)\} + \|z(t)\|$, which is a difficult condition to verify.

*B. Stability Theorem of DDSs*

Next, a lemma that can be used to realize synchronization of CDDSs with intermittent control is provided.

*Lemma 2:* Given a time sequence $\{t_\rho\}_{\rho=0}^{\infty}$ with $t_0 = 0$, $\lim_{\rho \to +\infty} t_\rho = +\infty$, and $\lim_{\rho \to +\infty} \sup \frac{t_{2\rho+2} - t_{2\rho+1}}{t_{2\rho+2} - t_{2\rho}} = \phi \in (0, 1)$, if there is a continuous and nonnegative function $w(t)$ with $t \in [-\sigma, +\infty)$ such that

$$\begin{cases} \dot{w}(t) \leq -a_1 w(t) + b\overline{w}(t) - c_1, & t \in \mathfrak{c}_\rho = [t_{2\rho}, t_{2\rho+1}), \\ \dot{w}(t) \leq a_2 w(t) + b\overline{w}(t) + c_2, & t \in \mathfrak{u}_\rho = [t_{2\rho+1}, t_{2\rho+2}), \end{cases} \tag{9}$$

then it has that $w(t) < Me^{-\tilde{\lambda}t}$, $\tilde{\lambda} = \lambda - (a_1 + a_2)\phi > 0$, $t \geq 0$, where $\rho \in \mathbb{N}$, $M > 0$, $\overline{w}(t) = w(t - \sigma(t))$, $\lambda > 0$ is the unique solution of transcendental equation $a_1 - \lambda - b_2 e^{\lambda \sigma} = 0$, and the other parameters meet that $a_1 > b \geq 0$, $c_1 = (a_1 - b)d > 0$, and $c_2 = (a_2 + b)d > 0$.

*Proof:* Let $h(t) = w(t) + d$. Then, it has that $\overline{h}(t) = \overline{w}(t) + d$ and $h(s) = \phi(s) + d > 0$, $s \in [-h, 0]$,

$$\begin{cases} \dot{h}(t) \leq -a_1 h(t) + b\overline{h}(t), & t \in \mathfrak{c}_\rho, \\ \dot{h}(t) \leq a_2 h(t) + b\overline{h}(t), & t \in \mathfrak{u}_\rho, \end{cases} \tag{10}$$

Following the results of [14], it concludes from the definition of $h(t)$ and (10) that $w(t) < h(t) \leq \sup_{s \in [-\sigma, 0]} \overline{h}(s) e^{-\tilde{\lambda}t}$. By defining $M = \sup_{s \in [-\sigma, 0]} \overline{h}(s)$, the proof is finished. ∎

*C. Research Problem*

This talk discusses the complete synchronization of coupled networks with $\ell$ DDSs (1) via an event-triggered intermittent controller. The coupled network is modeled as

$$\begin{cases} \dot{x}_s(t) = F(x_s, x_{s,\sigma}) + \sum_{j=1}^{\ell} u_{sj} \Phi x_j(t) + r_s(t), \\ x_s(o) = \tau_s(o) \in \mathcal{C}([-\sigma, 0], \mathbb{R}^n), s \in \mathbb{N}_1^{\ell}, \end{cases} \tag{11}$$

where $x_s(t), r_s(t) \in \mathbb{R}^n$ are respectively the state variable and the control input, outer-coupling matrix $U = (u_{ij})_{\ell \times \ell}$ satisfies the diffusive condition, $\Phi$ is the inner-coupling matrix. Similar to (2), the CDDSs (11) in sense of Filippov solution is

$$\dot{x}_s(t) = \mathbb{F}(x_s, \gamma_s, \zeta_{s,\sigma}) + \sum_{j=1}^{\ell} u_{sj} \Phi x_j(t) + r_s(t), \tag{12}$$

where $\mathbb{F}(x_s, \gamma_s, \zeta_{s,\sigma}) = Cx_s(t) + A\gamma_s(t) + B\zeta_s(t - \sigma(t))$, $\gamma_s(t) \in \mathsf{F}\{h(x_s(t))\}$ and $\zeta_s(t - \sigma(t)) \in \mathsf{F}\{g(x_s(t - \sigma(t)))\}$.

*Definition 1:* The CDDSs (11) is said to be globally exponentially synchronized with DDS (1) if, by designing suitable controllers $r_s(t), s \in \mathbb{N}_1^{\ell}$, there exist $M \geq 0$ and $\alpha > 0$ such that $\|\mathsf{e}(t)\| \leq Me^{-\alpha t}$, for $t \geq 0$, where $\mathsf{e}(t) = \mathrm{cl}(e_s(t))_{\ell}$, $e_s(t) = x_s(t) - z(t)$.

## III. SYNCHRONIZATION OF CDDSs

*A. Control Design*

According to [8], the control goal presented in Definition 1 is equivalence to the same issue for the Filippov systems (2) and (12). Hence, the subsequent study directly addresses the synchronization issue of (2) and (12). In this talk, the new event-triggered intermittent control is designed as

$$r_s(t) = \begin{cases} -K_s e_s(t_k^{s,2\rho}) - \xi_s \mathsf{sg}(e_s(t_k^{s,2\rho})), \\ \quad t \in \mathfrak{c}_\rho \cap [t_k^{s,2\rho}, t_{k+1}^{s,2\rho}), \\ 0, \ t \in \mathfrak{u}_\rho, \end{cases} \tag{13}$$

where $\xi_s > 0$ and $K_s \in \mathbb{R}^{n \times n}$ are the control gains, $t_k^{s,2\rho}$ is the $k^{th}$ control signal update instant of subsystem $s$, which is determined by the following ETM

$$t_{k+1}^{s,2\rho} = \inf\{t > t_k^{s,2\rho} : \|\theta_s(t)\| - \kappa_s \|e_s(t_k^{s,2\rho})\| > 0\}, \tag{14}$$

where $t_0^{s,2\rho} = t_{2\rho}$, $\theta_s(t) = e_s(t_k^{s,2\rho}) - e_s(t)$ is the ME and $\kappa_s \in (0, 1)$ is the threshold value.

*Remark 2:* The ME $\theta_s(t)$ in (14) is linear and demands less computing power than the nonlinear ones, such as those in [11], [12], [17], which will further be clarified in the numerical example part. In addition, it observes that the MEs in [11], [12], [17] are piecewise continuous, which also introduce additional challenges in proving the exclusion of Zeno behavior. While, these challenges will not arise in the case of a linear ME. Hence, event-triggered nonsmooth control with a linear ME is more practical.

Considering system (2) and CDDSs (12) with controller (13), the error system is obtained as

$$\dot{e}_s(t) = \mathbb{F}_s(t), \ t \in \mathfrak{c}_\rho, \tag{15a}$$

$$\dot{e}_s(t) = \widetilde{\mathbb{F}}_s(t), \ t \in \mathfrak{u}_\rho, \ \rho \in \mathbb{N}, \tag{15b}$$

and its compact Kronecker product form is

$$\dot{\mathsf{e}}(t) = \mathsf{F}(\mathsf{e}, \theta, \mathsf{r}, \mathsf{c}_\sigma), \ t \in \mathfrak{c}_\rho, \tag{16a}$$

$$\dot{\mathsf{e}}(t) = \widetilde{\mathsf{F}}(\mathsf{e}, \theta, \mathsf{r}, \mathsf{c}_\sigma), \ t \in \mathfrak{u}_\rho, \ \rho \in \mathbb{N}, \tag{16b}$$

where $\mathbb{F}_s(t) = \widetilde{\mathbb{F}}_s(t) - \xi_s \mathsf{sg}(e_s(t) + \theta_s(t)) - K_s(e_s(t) + \theta_s(t))$, $\widetilde{\mathbb{F}}_s(t) = Ce_s(t) + Ar_s(t) + Bc_s(t - \sigma(t)) + \sum_{j=1}^{\ell} u_{sj} \Phi e_j(t)$, $\mathsf{F}(\mathsf{e}, \theta, \mathsf{r}, \mathsf{c}_\sigma) = \widetilde{\mathsf{F}}(\mathsf{e}, \theta, \mathsf{r}, \mathsf{c}_\sigma) - \mathcal{K}(\mathsf{e}(t) + \theta(t)) - \xi\mathsf{sg}(\mathsf{e}(t) + \theta(t))$, $\widetilde{\mathsf{F}}(\mathsf{e}, \theta, \mathsf{r}, \mathsf{c}_\sigma) = (\mathcal{C} + \mathcal{U})\mathsf{e}(t) + \mathcal{A}\mathsf{r}(t) + \mathcal{B}\mathsf{c}(t - \sigma(t))$, $\theta(t) = \mathrm{cl}(\theta_s(t))_{\ell}$, $\mathsf{r}(t) = \mathrm{cl}(\mathsf{r}_s(t))_{\ell}$, $\mathsf{r}_s(t) = \gamma_s(t) - \gamma(t)$, $\mathsf{sg}(\mathsf{e}(t) + \theta(t)) = \mathrm{cl}(\mathsf{sg}(e_s(t) + \theta_s(t)))_{\ell}$, $\mathsf{c}(t - \sigma(t)) = \mathrm{cl}(\mathsf{c}_s(t - \sigma(t)))_{\ell}$, $\mathsf{c}_s(t - \sigma(t)) = \zeta_s(t - \sigma(t)) - \zeta(t - \sigma(t))$ $\mathcal{X} = I_{\ell} \otimes X$, $X \in \{A, B, C\}$, $\mathcal{U} = U \otimes \Phi$, $\mathcal{K} = \mathsf{dg}(K_s)_{\ell}$, and $\xi = \mathsf{dg}(\xi_s I_n)_{\ell}$.

## B. Synchronization Analysis

The synchronization criteria are given below.

*Theorem 1:* Assume that $(\mathbf{A}_1)$ holds. For given $\phi, \kappa_s \in (0,1)$, $a_1 > b = \|\mathcal{B}_D^g\|$, and $a_1 + a_2 > 0$, there are matrices $\mathcal{K} = \mathsf{dg}(K_s)_\ell \in \mathbb{R}^{\ell n \times \ell n}$ and $\Psi = \mathsf{dg}(\Psi_s)_\ell \in \mathbb{D}_+^{\ell n \times \ell n}$ such that $\eta = \frac{a_1 - b}{a_2 + b}\upsilon > 0$, $\zeta_s = \frac{1 + \widetilde{\kappa}_s}{1 - \widetilde{\kappa}_s}\eta, \xi_s = \frac{1 + \widetilde{\kappa}_s}{1 - \widetilde{\kappa}_s}\upsilon + \zeta_s, s \in \mathbb{N}_1^\ell$,

$$\Omega_1 = \begin{pmatrix} \mathsf{He}[\mathbb{A}_1 + \mathcal{A}_D^h] + \widetilde{\Psi} & -\mathcal{K} \\ * & -\Psi \end{pmatrix} < 0, \qquad (17)$$

$$\Omega_2 = \mathsf{He}[\mathbb{A}_2 + \mathcal{A}_D^h] < 0, \qquad (18)$$

then CDDS (11) with controller (13) is globally exponentially synchronized onto DDS (1), i.e., $\|\mathsf{e}(t)\| \le M e^{-\tilde{c}t}$, $\tilde{c} = c - (a_1 + a_2)\phi > 0$, where $c$ is the solution of $a_1 - c - be^{c\sigma} = 0$, $\phi$ is defined in Lemma 2, $M = \sup_{s \in [-\sigma, 0]} \|\mathsf{e}(s)\| + \frac{\upsilon}{a_2 + b}$, $\mathbb{A}_1 = \mathcal{C} - \mathcal{K} + \mathcal{U} + a_1 I_{\ell n}$, $\mathbb{A}_2 = \mathcal{C} + \mathcal{U} - a_2 I_{\ell n}$, $\widetilde{\Psi} = \mathsf{dg}(\widetilde{\kappa}_s^2 \Psi_s)_\ell$, $\mathcal{A}_D^h = I_\ell \otimes (\sum_{r=1}^n |a_{ir}| d_{rj}^h)_{n \times n}$, $\mathcal{B}_D^g = I_\ell \otimes (\sum_{r=1}^n |b_{ir}| d_{rj}^g)_{n \times n}$, $\mathsf{a}_h = \ell^{\frac{1}{2}} \|\mathsf{cl}(\sum_{r=1}^n |a_{ir}| \widehat{d}_r^h)_n\|$, $\mathsf{b}_g = \ell^{\frac{1}{2}} \|\mathsf{cl}(\sum_{r=1}^n |b_{ir}| \widehat{d}_r^g)_n\|$, $\widetilde{\kappa}_s = \frac{\kappa_s}{1 - \kappa_s}$, and $\upsilon = \mathsf{a}_h + \mathsf{b}_g$.

*Proof:* Design a Lyapunov function $V(t) = \|\mathsf{e}(t)\|$.
For $t \in \mathsf{c}_\rho$, $\rho \in \mathbb{N}$, it derives from (16a) that

$$\mathcal{D}^+[V(t)] = \frac{2\mathsf{e}^\mathsf{T}(t)\mathsf{F}(\mathsf{e}, \theta, \mathsf{r}, \mathsf{c}_\sigma)}{2V(t)}. \qquad (19)$$

It follows from $(\mathbf{A}_1)$ and Cauchy-Schwarz inequality that

$$\mathsf{e}^\mathsf{T}(t)\mathcal{A}\mathsf{r}(t) \le \mathsf{e}^\mathsf{T}(t)\mathcal{A}_D^h \mathsf{e}(t) + \mathsf{a}_h \|\mathsf{e}(t)\|, \qquad (20)$$

$$\mathsf{e}^\mathsf{T}(t)\mathcal{B}\mathsf{c}(t - \sigma(t)) \le (b\|\mathsf{e}(t - \sigma(t))\| + \mathsf{b}_h)\|\mathsf{e}(t)\|. \qquad (21)$$

The ETM (14) means $\|\theta_s(t)\| \le \widetilde{\kappa}_s \|e_s(t)\|$ and

$$\theta^\mathsf{T}(t)\Psi\theta(t) \le \mathsf{e}^\mathsf{T}(t)\widetilde{\Psi}\mathsf{e}(t). \qquad (22)$$

Moreover, one has from $\|\theta_s(t)\| \le \widetilde{\kappa}_s \|e_s(t)\|$ that

$$\begin{aligned} \mathsf{e}^\mathsf{T}(t)\xi\mathsf{sg}(\mathsf{e}(t) + \theta(t)) &\ge \sum_{s=1}^\ell \frac{\xi_s \|e_s(t)\|(\|e_s(t)\| - \|\theta_s(t)\|)}{\|e_s(t) + \theta_s(t)\|} \\ &\ge \sum_{s=1}^\ell \frac{\xi_s(1 - \widetilde{\kappa}_s)\|e_s(t)\|^2}{(1 + \widetilde{\kappa}_s)\|e_s(t)\|} \\ &\ge (\upsilon + \eta)\|\mathsf{e}(t)\|. \end{aligned} \qquad (23)$$

Substituting inequalities (20)–(23) into (19) yields

$$\begin{aligned} \mathcal{D}^+[V(t)] \le &\frac{\varepsilon^\mathsf{T}(t)\Omega\varepsilon(t) + 2bV(t)V(t - \sigma(t))}{2V(t)} \\ &- a_1 V(t) - \eta, \end{aligned} \qquad (24)$$

where $\varepsilon(t) = (\mathsf{e}^\mathsf{T}(t), \theta^\mathsf{T}(t))^\mathsf{T}$. Then, condition (17) and inequality (24) ensure that

$$\mathcal{D}^+[V(t)] \le -a_1 V(t) + bV(t - \sigma(t)) - \eta. \qquad (25)$$

Similarly, for $t \in \mathsf{u}_\rho$, $\rho \in \mathbb{N}$, it has from (16b) and (18) that

$$\mathcal{D}^+[V(t)] \le a_2 V(t) + bV(t - \sigma(t)) + \upsilon. \qquad (26)$$

Then, from Lemma 2 and inequalities (25)–(26), the result of Theorem 1 can be obtained. ∎

*Remark 3:* Based on the novel nonsmooth event-triggered intermittent control (13) and Lemma 2, Theorem 1 presents the complete synchronization criteria for CDDS (11). The result is quite general since Theorem 1 allows that the derivative of $\sigma(t)$ is less, equal to, greater than 1, or even that $\sigma(t)$ is nondifferentiable. Specially, when the derivative of the delay $\sigma(t)$ exceeds 1 or even delay $\sigma(t)$ is nondifferentiable, the nonsmooth control (13) makes the Lyapunov-Krasovskii functional methods to show limitations in achieving the complete synchronization. The main reason is that many techniques dealing with time delay in the Lyapunov-Krasovskii functional methods only depend on linear controls, which cannot achieve the complete synchronization of CDDS (11). Hence, a new analysis framework of studying the complete synchronization of CDDSs with intermittent control is proposed.

Next, let us discuss the Zeno behavior of ETM (14).

*Theorem 2:* Under the assumption and conditions of Theorem 1 the triggering instants generated by ETM (14) can rule out the Zeno behavior.

*Proof:* For $\forall s \in \mathbb{N}_1^\ell$ and $t \in \mathsf{c}_\rho \cap [t_k^{s,2\rho}, t_{k+1}^{s,2\rho})$, it has that

$$\mathcal{D}^+[\|\theta_s(t)\|] \le \|\mathcal{D}^+[e_s(t_k^{s,2\rho}) - e_s(t)]\| = \|\dot{e}_s(t)\|. \qquad (27)$$

In view of Theorem 1, it concludes that there is a $\mathsf{u}_s > 0$ such that $\|e_s(t)\| \le \mathsf{u}_s$. Then, one can obtain from error system (15a), and $(\mathbf{A}_1)$ that

$$\|\dot{e}_s(t)\| \le \vartheta_s + \|K_s\|\|\theta_s(t)\|, \qquad (28)$$

where $\vartheta_s = (\|C - K_s\| + \|A_D^h\| + \|B_D^g\|)\mathsf{u}_s + \upsilon + \xi_s + 2|u_{ss}|\|\Phi\|\sum_{j=1}^\ell \mathsf{u}_j$, $A_D^h = (\sum_{r=1}^n |a_{ir}| d_{rj}^h)_{n \times n}$, and $B_D^g = (\sum_{r=1}^n |b_{ir}| d_{rj}^g)_{n \times n}$.

One has from inequalities (27)–(28) and $\|\theta_s(t_k^{s,2\rho})\| = 0$ that $\|\theta_s(t)\| \le \frac{\|K_s\|}{\vartheta_s}(e^{\|K_s\|(t - t_k^{s,2\rho})} - 1)$, that is, $(t - t_k^{s,2\rho}) \ge \frac{1}{\|K_s\|} \ln(\frac{\|K_s\|}{\vartheta_s}\|\theta_s(t)\| + 1)$. Note that, the next event will not be triggering until $\|\theta_s(t_{k+1}^{s,2\rho-})\| = \kappa_s \|e_s(t_k^{s,2\rho})\|$. Hence, the inequality above implies that $(t_{k+1}^{s,2\rho-} - t_k^{s,2\rho}) \ge \frac{\ln(\frac{\|K_s\|\kappa_s}{\vartheta_s}\|e_s(t_k^{s,2\rho})\| + 1)}{\|K_s\|} > 0$. ∎

## IV. NUMERICAL EXAMPLE

This section utilizes the Hopfield neural network (HNN) with discontinuous activation functions to verify the effectiveness of our results. The circuit diagram of the HNN is shown in Fig. 1(a) with detailed explanations provided in [23]. By applying Kirchhoff's laws, the HNN can be represented as a DDS (1). Next, the parameters of the HNN, in the form of those in DDS (1), are selected for numerical simulation.

Conside a HNN or the DDS (1) with $z(t) = (z_1(t), z_2(t))^\mathsf{T}$, $g(z) = (g_1(z_1), g_2(z_2))^\mathsf{T}$, $h(z) = (h_1(z_1), h_2(z_2))^\mathsf{T}$, $\sigma(t) = 0.65 + 0.35|\sin(t)|$, $C = \mathsf{dg}(-1.5, -1)$, $i = 1, 2$,

$$A = \begin{pmatrix} 2 & -0.1 \\ -4.9 & 3 \end{pmatrix}, g_i(z_i) = \begin{cases} \frac{|z_i + 1| - |z_i - 1|}{2} + 0.04, z_i > 0, \\ \frac{|z_i + 1| - |z_i - 1|}{2} - 0.01, z_i < 0, \end{cases}$$

$$B = \begin{pmatrix} -1.5 & 0.1 \\ -0.5 & -0.5 \end{pmatrix}, h_i(z_i) = \begin{cases} \tanh(z_i) + 0.01, z_i > 0, \\ \tanh(z_i) - 0.02, z_i < 0. \end{cases}$$

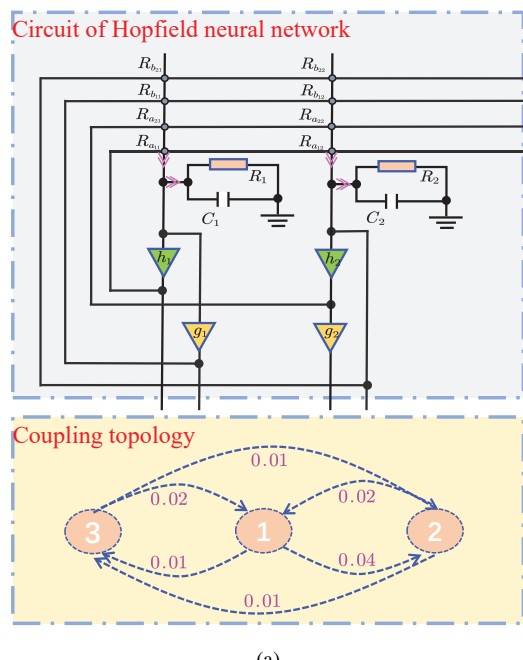

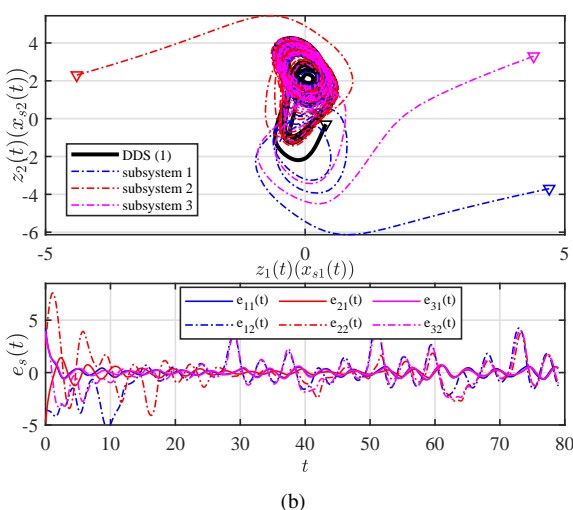

Fig. 1: (a) Circuit diagram of the HNN and coupling topology; (b) Trajectories of DDS (1) and CDDS (11) without controller.

It has that $\mathbb{a}(\cdot), \mathbb{a} = \{h, g\}$ meet $(\mathbf{A}_1)$ with $d_{11}^{\mathbb{a}} = d_{22}^{\mathbb{a}} = 1$, $d_{12}^{\mathbb{a}} = d_{21}^{\mathbb{a}} = 0$, $\hat{d}_1^h = \hat{d}_2^h = 0.03$, and $\hat{d}_{21}^g = \hat{d}_{21}^g = 0.05$.

Now, consider that the coupled system (11) is composed of 3 DDS (1), where $\Phi = \mathrm{dg}(2, 1)$ and $U = (u_{ij})_{3\times3}$ is the Laplacian matrix of the digraph shown in Fig. 1(a). When the initial values of DDS (1) and CDDS (11) are randomly chosen on $[-5, 5]$, $\forall t \in [-1, 0]$, their trajectories are given in Fig. 1(b), from which one can see that the synchronization cannot be realized without the control.

By taken $a_1 = 4.6, a_2 = 3.88, \kappa_1 = 0.12, \kappa_2 = 0.17$, and $\kappa_3 = 0.15$, one gains that $b = 1.603$ $\xi_1 = 1.197$, $\xi_2 = 1.378$, $\xi_3 = 1.299$ and $\phi = 0.1002$. Solving conditions

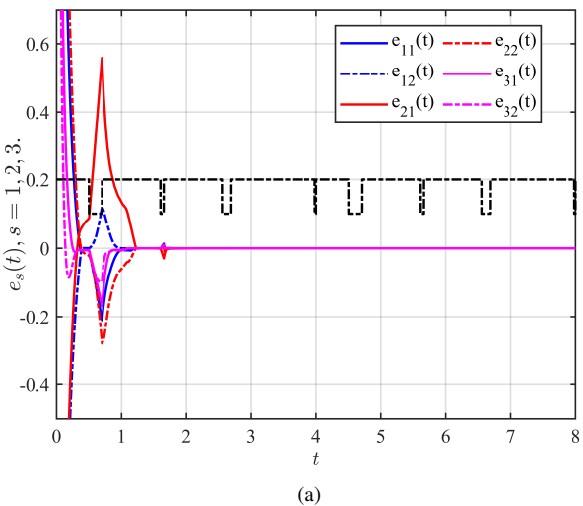

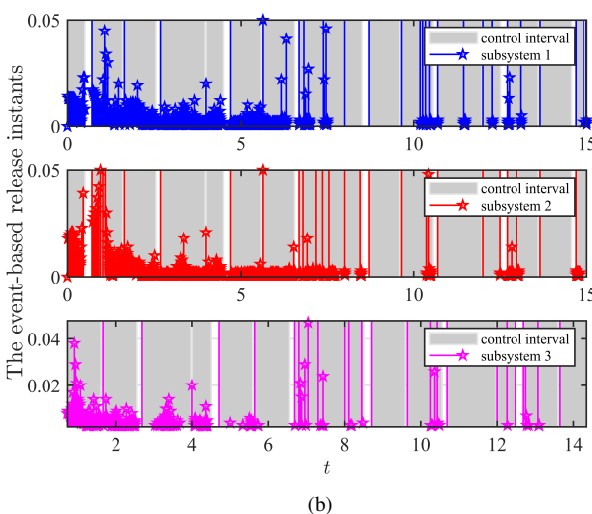

Fig. 2: (a) Error trajectories of DDS (1) and CDDS (11) with controller (13); (b) Triggering instants and intervals.

(17) and (18) obtains $K_1 = \begin{pmatrix} 11.480 & 3.759 \\ 3.759 & 13.908 \end{pmatrix}$, $K_2 = \begin{pmatrix} 11.690 & 3.815 \\ 3.815 & 14.139 \end{pmatrix}$, $K_3 = \begin{pmatrix} 11.744 & 3.854 \\ 3.854 & 14.236 \end{pmatrix}$. Hence, Theorem 1 is true, that is, CDDS (11) with controller (13) can be synchronized onto DDS (1). Fig. 2(a) shows the evolution of error trajectories of (11) and (1) when the work intervals of controller (13) are $[0, 0.5) \cup [0.5, 0.7) \cup [0.7, 1.6) \cup [1.6, 1.65) \cup [1.65, 2.55) \cup [2.55, 2.68) \cup [2.68, 3.98) \cup [3.98, 4) \cdots$. In addition, the triggering instants and intervals of three subsystems are displayed in Fig. 2(b), respectively. It finds from Fig. 1 (b) and Fig. 2 that the designed event-triggered controller (13) is not only efficient but also resource-efficient.

**Comparative Experiment:** To prove the novelty **3)**, a comparative experiment with the ETMs from in [11], [12], [17] is conducted, where average running time (**ART**) and trigger rate (**RT**) are the measurement standards. The results are listed

TABLE I: **TR**[1] and **ART**[2] of ETM (14) and [11], [12], [17].

| Methods | (14) | | | [11], [12], [17] | | |
|---|---|---|---|---|---|---|
| Nodes | 1 | 2 | 3 | 1 | 2 | 3 |
| **TR** (%) | **27.17** | **36.43** | **31.84** | 39.51 | 38.93 | 38.38 |
| **ART** (sec) | **0.5214** | | | 0.7966 | | |

[1]**TR**= $\frac{\text{The number of trigger releases}}{\text{Total signals}}$ ; [2]**ART** is the average obtained from 10 runs of the code.

in Table I. In the simulation, the time-step size is 0.001, and a total of 12420 control signals are generated for $[0, 15]$. The experiment code runs on a computer with Windows 10, Intel Core i5-10400, 2.9GHz, and 16GB RAM. It observes from Table I that ETM (14) not only saves $52.78\%$ of the running time but also reduces trigger frequency.

## V. CONCLUSION

This talk has considered the complete synchronization of CDDSs under event-triggered intermittent control. By developing a new stability inequality and a weighted-norm-based Lyapunov function, sufficient synchronization conditions have been derived. Note that, the results of this talk did not impose any restrictions on the derivatives of the delay. Moreover, experiments shown that the novel event-triggered control with a linear ME requires less computing power than existing papers.

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
