# OpenReview forum: "Synchronization of Coupled Delayed Discontinuous Systems via Event-Trigged Intermittent Control"
_IEEE.org/ICIST/2024/Conference — IEEE ICIST 2024 Conference Submission_

### Official Review · Reviewer_MNWi · 2024-08-23
**This article mainly accomplishes the development of a novel control scheme for achieving complete synchronization of coupled delayed discontinuous systems (CDDSs). The authors propose an event-triggered intermittent control method that effectively synchronizes CDDSs with reduced computational burden.**

**Rating:** 8
**Confidence:** 4

**Review:**

This article mainly accomplishes the development of a novel control scheme for achieving complete synchronization of coupled delayed discontinuous systems (CDDSs). The authors propose an event-triggered intermittent control method that effectively synchronizes CDDSs with reduced computational burden. In general, this work is well organized and appears potentially interesting, it can be accepted with a little modification.
1.	Improper use of articles or long sentence structures with separators might make it difficult to follow the paper. For example, the use of ‘the’ in the paper and the presentation of subordinate clauses. Please check the full text again and modify the grammar problems.
2.	How does your paper significantly go beyond existing results?
3.	Please highlight the contributions of the paper.
4.	Please add the necessary comments for Figures.

---

### Official Review · Reviewer_mEHW · 2024-08-23
**accept**

**Rating:** 7
**Confidence:** 3

**Review:**

This paper presents the complete synchronization of a coupled delayed discrete-time non-autonomous system, which demonstrated excellent performance. The theory is correct and can be accepted after responding the following comments.
(1)In the introduction, it is not enough to state the current work. It should be expended and reconstructed.
(2)Please check if you need to update your Introduction.
(3)In the simulation section, more analysis can be added to better explain the main results of this paper, that's not enough.
(4)There are many typos and grammar errors. The authors should have a native English speaker or software packages to perform the editing check.

---

### Official Review · Reviewer_tdsm · 2024-08-24
**This paper introduced the complete synchronization of coupled delayed discontinuous systems (DDSs). Without constraints on the derivatives of time delays, several new conditions are exploited to guarantee the global existence of Filippov solutions for DDSs. A nonsmooth intermittent control combined with an event-triggering strategy is then designed. The topic of this paper is interesting. Below is a list of comments that should be taken into account further when revising the paper.**

**Rating:** 7
**Confidence:** 3

**Review:**

1. In the preliminaries section, the author should provide readers with a detailed description of the control objectives of this article, so that they can clearly understand the ultimate goal of this article.
2. In the simulation results section, the horizontal axis time of Figure 1 and Figure 2 should be longer, which ensures that readers can have a more intuitive understanding of the advantages of this method.
3. The paper should provide a detailed description of the innovative points to enable readers to quickly understand the article. Meanwhile, please elaborate on the future plans.

---

### Decision · Program_Chairs · 2024-09-06

Accept (Oral)